# Analysing factors influencing railway accidents: A predictive approach using multinomial logistic regression and data mining

Jaroslav Mašek[1]*, Lucia Duricova[2], Juraj Čamaj[1]

1 Department of Railway Transport, Faculty of Operation and Economics of Transport and Communications, University of Zilina, Zilina, Slovak Republic, 2 Department of Economics, Faculty of Operation and Economics of Transport and Communications, University of Zilina, Zilina, Slovak Republic

* jaroslav.masek@uniza.sk

## Abstract

Railway accidents, particularly suicides and suicide attempts, significantly disrupt operations, cause delays in passenger and freight services, and result in varying degrees of infrastructure damage. This study focuses on identifying the relationship between suicide-related railway incidents, as the most frequent type of railway accidents, and socio-economic factors, utilising data from 2015 to 2022 provided by the Railways of the Slovak Republic. Using a data mining approach, a logistic regression model was developed to predict the accident rate based on key socio-economic factors. This model demonstrates high prediction performance, with significant predictors including interest, marriage, and fertility rates. The data mining approach allows for the efficient extraction of relevant patterns and relationships and ensures that the model can be easily adjusted in response to significant changes in input factors or conditions. The findings contribute to understanding railway safety, offering practical insights for improving safety measures and aiding suicide prevention efforts. The high explanatory power of the predictive model underscores the critical role of societal influences in the dynamics of railway-related suicides and suicide attempts, providing valuable guidance for enhancing safety protocols and planning.

## 1. Introduction

The safety of railway transport is of high importance, with railway accidents posing significant challenges worldwide [1]. Such events have a negative impact not only on the quality of infrastructure but also on the quality of the transport process in passenger and freight transport. Factors such as unemployment rates, demographic trends, and mental health issues can influence accident occurrence and severity. Understanding the relationship between the accidents and their potential causal factors is crucial for implementing effective preventive measures to enhance railway safety. On the other hand, predictive modelling and machine learning techniques offer a very

**Data availability statement:** All relevant data are within the manuscript and its Supporting Information files.

**Funding:** The paper was supported by the KEGA Agency by Project 010ŽU-4/2023. The funders had no role in study design, data collection and analysis, decision to publish, or preparation of the manuscript.

**Competing interests:** The authors have declared that no competing interests exist.

usable approach for analysing historical data, providing valuable insights into hidden trends, patterns and relationships. Thus, a novel approach to the field of analysis of railway accidents could be brought by using predictive modelling techniques, which offer valuable insights into accident prediction and prevention strategies [2].

This paper aims to contribute to the current state of the art by bridging the gap between traditional accident analysis and predictive modelling and suggesting possible solutions to prevent such events. The paper aims to connect three main areas: railway accident causation, predictive modelling approaches, and the influence of societal factors on railway safety. We consider this study important for several reasons. Firstly, it addresses a critical gap in understanding the interplay between societal factors and railway accidents, an area that has been underexplored in existing literature, as described here below within the literature review. By leveraging comprehensive data covering a period of eight years, the research offers robust insights into accident causation, which is vital for developing effective safety strategies. We see the contribution of the mentioned connection of the three areas in the creation of an accurate model that will be able to estimate the number of accidents with a high degree of reliability with known values of the input factors that will be the explanatory variables of the model. Therefore, the study's novelty lies in integrating predictive modelling, a forward-looking approach that not only analyses past incidents but also forecasts future risks based on societal trends. This predictive capability is crucial for proactive safety management and can be adapted by railway systems worldwide. International readers will find the study's methodologies and findings applicable across different geographical contexts, providing a valuable framework for enhancing railway safety globally. The implications for suicide prevention further underscore the study's societal impact, offering actionable insights for reducing fatalities on rail networks.

The rest of the paper is organised as follows. First, the literature review section highlights existing studies on railway safety and accident causation. Following this, the theoretical research background section provides a theoretical framework. The methodology and data section briefly describes the research approach and modelling techniques. The results section presents the main findings and describes and evaluates the prediction model. In the discussion section, the results are interpreted in the context of existing literature, highlighting key implications. Finally, the conclusion section summarises the main findings and possible future directions of the study.

## 2. Literature review

Railway accidents pose significant challenges to transportation safety and infrastructure worldwide. Understanding the factors contributing to these accidents and implementing effective preventive measures is crucial for ensuring the safety of railway transport systems. In his study, Schlesinger [3] stated that although travel by train is the safest mode of transport in the US, in case of major accidents, it is crucial to discuss the aspects of the accident and implement measures resulting from the findings.

In literature, various theories and models exist to understand the causation of accidents. The accident Causation Model, known as the Swiss Cheese Model, provides

insights into the multiple layers of factors that contribute to accidents, such as organisational, human, and technical aspects [4,5,6]. Additionally, some studies have investigated the role of human factors, organisational culture, and safety management systems in accident causation within railway systems (e.g., [7,8]). Mathew et al. [9] stated that safety at railroad grade crossings is critical for rail networks. The authors suggested an automated method for sorting and clustering accident attributes to identify the most important accident factors. The accidents at railway level crossings are also investigated in a study by Evans and Hughes [10]. The authors revealed the relationships between fatalities at these level crossing and traverses and delays.

In this study, we aimed at the railway accidents of suicide or suicide attempt type and their relationship with the socio-economic indicators as potential – direct or indirect – causation factors. The connection between suicide rates and the socio-economic environment in general has been well discovered in the literature. Extensive research in this area has uncovered a significant correlation between socio-economic indicators, such as low income, unemployment, and financial difficulties, and suicide rates. These factors have consistently been identified as risk factors for suicidal behaviour. In the study by Raschke et al. [11], the authors collected 35 studies for a systematic review focused on the association between individual socio-economic factors and suicidal ideation. They identified several factors, including low income, unemployment, and financial difficulties, as risk factors for all suicidal behaviours. Milner et al. [12] also confirm the relationship between socio-economic environment and suicide rates in general. However, in this study, the authors aimed at the association between socio-economic variables and gender-specific suicide rates in 35 countries while controlling for country-specific influences. Their study offers detailed results on the association of suicide rates with increased female labour force participation, unemployment, and the proportion of older people in the population. On the other hand, lower gender-specific suicide rates were associated with increased health spending per capita, and higher fertility resulted in a reduction in male suicide. Furthermore, a study by Lorant et al. [13] across ten European countries found socio-economic inequalities in suicide mortality among men, underscoring the impact of socioeconomic disparities on mental health outcomes. These and many other studies confirm that the association between suicide rates and socio-economic factors is sufficiently discovered in the literature.

However, focusing solely on traffic accidents and their connection to socio-economic indicators, we have found that most existing studies are conducted on road transport. Nanjunda [14] examined the association between low socio-economic status and road accidents, particularly among disadvantaged youths. The results highlighted that socio-economic indicators are key factors contributing to road accidents in this demographic. Sohaee and Bohluli [15] emphasise the impact of, besides others, the socio-economic factors on the fatal traffic accident rate. The findings reveal that certain factors, such as the unemployment rate or minimum wage, among others, significantly impact the frequency of fatal road accidents. Majed et al. [16] investigated the relationship between economic growth and road fatalities in Iran and found a significant inverted U-shaped relationship of firstly increasing road accidents to a certain threshold income level and then decreasing with a further increase in income.

In contrast to road safety research, the influence of socio-economic factors on railway accident rates remains underexplored. Although there is a limited number of studies, they have predominantly concentrated on the technical, human, and organisational causes of train accidents [17]. While some research has assessed the socio-economic impacts of incidents in railway infrastructure [18], these studies often focus on the consequences rather than the underlying socio-economic factors. On the other hand, several studies investigated railway suicide or suicide attempts from the personal victim's viewpoint, trying to find causation factors for such accidents. Too et al. [19] collected eleven studies for their systematic review and found that certain population characteristics, such as population density or the presence of high-risk populations, were studied in them as causal factors of higher accident rates. Krysinska and Leo [20] primarily focused their analysis on the prevention of railway accidents, but they also noted that young adults are the group most at risk for railway suicides.

According to our knowledge, the overall impact of socio-economic factors on the railway accident rate is rather underexplored in the literature. This gap highlights the need for comprehensive investigations into how the socio-economic environment influences railway safety to inform targeted interventions and policy development.

 

Another point of view of railway accident analysis is their prediction. In recent years, there has been growing interest in applying predictive modelling techniques to predict and, thus, prevent railway accidents. Machine learning algorithms, such as logistic regression, decision trees, random forests, and neural networks, have been employed to analyse historical accident data and identify patterns or trends that can help predict future accidents [21,22,23]. These models not only quantify the likelihood of accidents under different input conditions but also offer the advantage of identifying potential risk factors.

Beyond technical and organisational factors, societal dynamics play a significant role in railway accidents. Some studies have explored the relationship between various societal factors, such as economic conditions, demographic trends, mental health indicators, and the occurrence of railway accidents [24,25,26].

Several years ago, Dindar et al. [27] presented the mathematical hierarchical Bayesian model, combining mathematical, statistical and geospatial approaches into a complex solution for predicting railway accidents or derailments. The Bayesian network was also used in the study by Xu and Xu [2] for predicting the risk of rail haulage accidents in mines. Prasetijo et al. [28] focused their study on finding the causes of railway accidents and modelling accidents using the Poisson regression model. The authors found the most significant factors influencing railway accidents.

Nowadays, machine learning techniques have come to the fore and have brought highly effective solutions for predicting accidents in railway transport. Mabrouk [29], in his study, proposed the use of machine learning and artificial intelligence techniques for improving the safety analysis methods in railway transport. The author proposed the tools serving two main purposes: recording and storing experiences and knowledge from safety analyses, and assisting those involved in evaluating safety studies.

Keramati et al. [30] proposed a random survival forest model for competing risks to investigate highway-rail grade crossing crash severity during a 29-year analysis period. The authors highlighted several advantages of the modelling technique used in their study. Moreover, as a result of the study, the most important predictors of each crash severity level were identified, and their marginal effects were evaluated. The authors quantified the effect of various preventive measures on crash likelihood. The random forest algorithm was also used in the study by Zhou et al. [31]. The authors compared the ensemble model created by the random forest algorithm with a simple decision tree, resulting in a significant improvement in model performance for predicting false alarms.

The study by Abioye et al. [32] provides an extensive review of the practice of identifying accident and hazard prediction over the years by departments of transportation in individual states. Furthermore, this study analyses the common factors usually used in accident and hazard prediction and identifies the most commonly used formula for predicting the number of accidents with higher accuracy.

Over the last seven years, research has shown that several authors have addressed accidents from different perspectives. However, the issue of railway safety, including research into causes and context, is very broad. Further investigation of this issue will help broaden the horizons of professionals and the public and bring a new perspective on this broad topic.

## 3. Theoretical background

In this study, we connect the areas of railway accident causation, predictive modelling approaches, and the influence of societal factors on railway safety. Therefore, this section briefly describes the background of the mentioned areas and is followed by the study methodology mentioning statistical and data analysis tools used in the study. Traditional accident causation theories have evolved to address the complexities of modern socio-technical systems. Heinrich's Domino Theory, or the Five Domino Model of accident causation, represents an accident resulting from a causal chain of events, represented as dominos. The fall of the first causes the fall of the second, followed by the third, etc. The author suggested that removing one factor in a domino sequence can prevent the accident [33]. However, this linear perspective has been critiqued for oversimplifying accident causation. Contemporary models, such as the Systems Theory Model,

view accidents as emergent properties of complex interactions within socio-technical systems, emphasising the interplay between human, technical, and environmental factors [34]. In the context of railways, accidents can stem from complex interactions among operators, technology, and passengers [35]. Effective risk management in railway operations involves systematic identification, assessment, and mitigation of potential hazards. The Swiss Cheese Model, introduced by Reason [5], illustrates how accidents occur due to aligned failures within organisational defences. Each slice of cheese represents a defensive layer, and holes symbolise weaknesses. When the holes align, an accident may occur. This model highlights the necessity for robust safety barriers and continuous monitoring to address vulnerabilities. In railway safety, implementing comprehensive risk assessment frameworks that account for human factors, technological reliability, and organisational policies is crucial [36].

Connecting the socio-economic environment to accidents could be vital in preventing them, as the outcomes of these analyses may indicate the potential factors influencing individuals to attempt suicide. Economic conditions, such as national income levels, unemployment rates, or inflation, can affect safety investments on one side and individual financial stability on the other, thereby impacting accident rates [36]. Additionally, societal aspects like marriage, divorce, or fertility rates can influence individuals' well-being and, therefore, may affect the railway accident rate. Understanding these socio-economic determinants is vital for developing targeted interventions and policies to enhance railway safety.

By integrating these theoretical perspectives, we can better understand the nature of railway accidents and develop more effective prevention and risk mitigation strategies.

## 4. Regulatory background

Railway safety is governed by various regulations aimed at enhancing both the safety and competitiveness of railways, particularly in relation to road transport. The fundamental railway safety legislation of the European Union is the Commission Directive [37] EC EU of 11th May 2016, upon which the safety regulations in the Slovak Republic are also based. This Directive applies to the railway system in all Member States, which may be divided into subsystems according to structural and functional areas. Its goals are the following:

• Harmonisation of the regulatory structure in the EU Member States,

• Defining the responsibilities of the various actors in the EU rail system,

• Establishment of common safety targets and common safety methods to progressively eliminate the need for national rules,

• Establishing principles for issuing, renewing, amending, and limiting or revoking safety certificates and authorisations,

• Requiring the setup of a national safety authority and an accident and incident investigation authority in each EU Member State,

• Establishing common principles for the management and regulation of railway safety and the supervision of railway safety.

In addition to this latest directive, two older directives can be mentioned: the 2014 Directive (2014/88/EC EU) and the 2004 Directive (2004/49/EC EU). Both of these have been incorporated into the aforementioned currently valid directive. Based on them, Act No. 513/2009 on Railways was adopted in Slovakia in 2009.

According to this Act, accidents are classified as serious, minor, and incidents involving a moving railway vehicle with consequences under paragraph 2 of this Act. On the railway network, regulation Z 17 is used for accident investigation.

According to paragraph 4, section 6 of this Act, it is prohibited to enter the railway and areas within the railway perimeter that are not open to the public without the consent of the railway operator. The exclusion zone includes, for example, platforms, crossings, and loading ramps. All the accidents analysed in this study occurred in areas that were not

accessible to the public, and none of them constituted an industrial accident; that is, the injured individuals were not working on the railway at the time of the incidents. Consequently, all the accidents examined were in violation of Act No. 513/2009 on Railways.

The most common accidents caused by the infrastructure manager include [38]:

• Faults on the track – inspections are conducted not only visually but also using technical devices installed on the track,

• Mistakes by employees – these occur especially with low-security operations (communication by phone call, mechanical device of switches),

• Failures of interlocking devices – the main cause is often the obsolescence of interlocking devices.

Significant factors in the accidents are external influences, such as weather conditions or the failure of road users to respect traffic light level crossings. Table 1 shows the distribution of accidents according to regulation Z 17, their consequences, and examples.

The table shows that accidents are divided into three groups according to consequences: loss of human lives, financial consequences, or damage to railway vehicle infrastructure. In this study, we focus only on accidents in categories A5 and B5, defined as injuries to persons caused by the movement of a railway vehicle [38].

To investigate accidents, types of injuries were categorised as follows [38]:

• Injury – damage to the health of persons caused by an external factor other than disease,

**Table 1. Accident categorisation.**

| Accident category | Consequence | Examples |
|---|---|---|
| A – serious accidents | Fatal injury | A1 - train collision |
| | Serious injury to at least five people | A2 - train derailment |
| | Extensive damage to rolling stock | A3 - collision of a railway vehicle with level-crossing users |
| | Extensive damage to railway infrastructure | A4 - railway vehicle fire |
| | Extensive environmental damage | A5 - personal injury caused by the movement of a railway vehicle |
| | Extensive damage to the property of third parties | |
| | Interruption of transport on tracks of the 1st category for at least 6 hours | |
| B – minor accidents | Serious injury to a maximum of four people | B1 - train collision |
| | Damage to railway vehicles and railway infrastructure | B2 - train derailment |
| | Damage to the environment and property of third parties to the extent of major damage of at least € 2,600 | B3 - collision of a railway vehicle with level-crossing users |
| | | B4 - railway vehicle fire |
| | | B5 - personal injury caused by the movement of a railway vehicle |
| C – incidents | Damage of a lesser extent | rail breakage |
| | | deformations of the rails |
| | | signalling error |
| | | overrunning of a signal in danger |
| | | wheel and axle breakage |

*Source:* [38]

- Fatal injury – any injury that causes the death of a person immediately or within 30 days of the accident if, according to medical opinion, the death resulted from the accident, excluding suicides,

- Serious injury – any injury to a person that results in hospitalisation for more than 24 hours, loss of an organ (anatomical or functional) or significant health damage, including industrial poisoning, described by the doctor as serious and caused by the accident, except suicide attempts,

- Suicide and suicide attempt – conduct leading to intentional self-injury with fatal or serious consequences, recorded and classified as such.

For determining the causes of accidents and for statistical purposes, persons affected by fatal and serious injuries are listed in Table 2.

## 5. Methodology and data

We aim to fulfil the main goal of this study by creating a predictive model for the number of railway accidents of suicide or suicide attempt types based on the knowledge of the input socio-economic and other contextual factors values. For this purpose, we follow the data-mining methodology CRISP-DM (Cross-Industry Standard Process for Data Mining) in this study, as the problem of predicting railway accidents should be considered a data mining task. According to this methodology, the process of solving this problem should be divided into six consecutive steps: *Business understanding*, *Data understanding*, *Data preparation*, *Modelling*, *Evaluation* and *Deployment*. Individual phases of the whole cycle are unidirectionally or bidirectionally connected. The whole process is depicted in Fig 1.

Therefore, the following parts of this article will correspond to these steps. The entire analysis was conducted using IBM SPSS Modeler software, versions 18.0 and 18.3, and some steps of the Data preparation phase were completed using IBM SPSS Statistics software, version 29.

Regarding the first step of *Business understanding*, we explained the need for predicting railway accidents and their purpose, along with understanding the influence of societal factors on railway safety, in the Introduction section and the necessary theoretical framework in the previous section.

The steps of *Data understanding* and *Data preparation* are closely connected; therefore, we describe them together in this section. For the purpose of predicting railway accidents, we used real data on railway accidents in the Slovak Republic from 2015 to 2022 provided by the railway infrastructure manager in Slovakia (Railways of the Slovak Republic). This accident database contains detailed information on all railway accidents, including their place and time, cause, type of train, gender and type of participating person and several other details.

Table 3 lists the number and proportions of accident types according to the accident cause. In this study, we further focus only on the most frequent accidents: suicides and suicide attempts.

**Table 2. Characterisation of accident participants with practical examples.**

| Kind of persons | Characteristic |
|---|---|
| passengers | persons (excluding train staff) travelling by train and persons attempting to board/get out to/ from a moving train in the boarding and alighting area |
| employers | persons involved in the management and provision of transport |
| users of railway crossing | persons using crossings by any means of transport or on foot |
| unauthorised persons in the railway precincts | persons without the appropriate permit located on the railway land and in the railway premises where such a stay is not allowed |
| others | persons not listed above, who are lawfully staying on the land and on the premises of Railways of the Slovak Republic |

*Source:* [38]

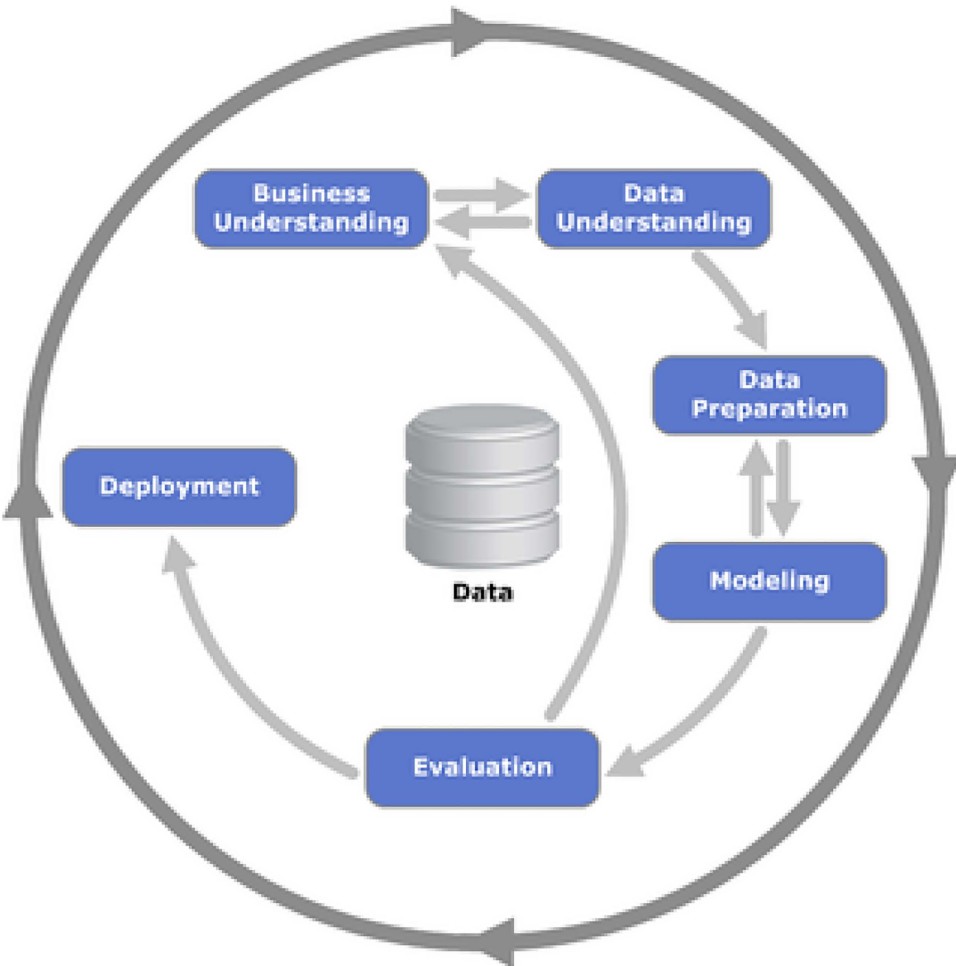

**Fig 1. Data mining process steps according to the methodology CRISP-DM.**

**Table 3. Number and shares of accidents according to their cause.**

| Accident cause | Proportion (%) | Count |
|---|---|---|
| Suicides and suicide attempts | 62.66 | 542 |
| Unauthorised movement of people in the cross-section of the track (outside the crossings) | 31.68 | 274 |
| Other causes | 5.43 | 47 |
| The cause is being investigated | 0.12 | 1 |
| Collision of a moving part and displacement between transport vehicles with a road vehicle, a pedestrian (cyclist), and transported animals outside a railway crossing | 0.12 | 1 |

For the purpose of this study, we aggregated the microdata on accidents into monthly totals for each month during the years under review. Thus, our target variable for creating the predictive model is the aggregated number of accidents using these two aggregating keys.

Next, we added several variables that could serve as appropriate explanatory factors for the number of accidents across different months of the year and regions of the country. All the explanatory factors, along with their description and data sources, are listed in Table 4.

To describe the database, we list the statistical characteristics of the quantitative explanatory variables and the outcome variable in Table 5 and the distribution of categories of qualitative explanatory variables in Table 6.

An interesting explanatory variable that could impact the number of accidents is the year of the accident. Table 6 shows the distribution of suicide or suicide attempt accidents during the years under review. The number of accidents peaked in 2022, with 83 fatalities on Slovak railways due to suicide or suicide attempts. In contrast, the lowest number was the year before, in 2021, with 54 incidents. Overall, the number of accidents appears relatively consistent across the years. In connection to the years under review, we can mention the distribution of accidents during the COVID-19 pandemic. In our study, the pandemic period covers the month froms March 2020 to February 2022. The aggregated number of accidents is indeed higher during the non-pandemic period (426 accidents in total), as it obviously covers a longer time span of 72 months, whereas the pandemic period lasted for 24 months (116 accidents total). However, the average monthly number of accidents during the pandemic period was 4.83, whereas for the non-pandemic period, it was 5.19. These two average numbers are not significantly distinct, but we will anyway consider the pandemic a possible explanatory factor in our prediction model.

Another perspective on the number of railway accidents is their distribution throughout the months of the year. These statistics help to identify the most critical months. Fig 2 details the distribution of suicide or suicide attempt accidents. The highest cumulative number of accidents occurred in November, totalling 64, followed by May with 57 accidents, while the lowest number, 29 accidents, was recorded in February. This suggests that the time of year significantly influences the number of railway accidents. Therefore, we created another explanatory variable, *season*, with the following values: 1 for spring months (March, April and May), 2 for summer months (June, July and August), 3 for autumn months (September, October and November) and 4 for winter months (December, January and February). This variable will be used alongside other explanatory variables in the prediction model. The distribution of accidents across the seasons is also presented in Fig 2. From this perspective, autumn is considered the most critical season, while winter is the least.

After preparing the database for predicting railway accidents, we can proceed to the next step of the data-mining process: *Modelling*. This step involves selecting a suitable machine learning technique to create a predictive model with high performance, together with the creation of the model itself. In the modelling phase, it is common to explore several modelling techniques suitable for the specific task. Therefore, we considered models such as linear regression, various classification trees (CART, CHAID, C5.0 methods), neural networks, and the nearest neighbour technique. It was found that these models achieve lower performance when predicting the precise number of accidents but higher performance when categorising the number of accidents into several categories instead of modelling it as a quantitative outcome. Therefore, we revisited the previous step of *Data preparation* and categorised the quantitative variable *number of accidents* into an ordinal variable *accident rate* with the following three categories:

- *Accident rate 1* (low accident rate): cumulative monthly accidents up to three,

- *Accident rate 2* (middle accident rate): cumulative monthly accidents from four to seven,

- *Accident rate 3* (high accident rate): cumulative monthly accidents of eight or more.

Through this process, we transformed the task of predicting the number of railway accidents from a regression task (where predicting a precise value of quantitative outcome variable is required) to a more common data mining classification task. Thus, the resulting predictive model predicts the category, or rather the accident rate, given the values of all explanatory variables. Categorising the outcome variable is a common practice in data mining and enhances the performance of the resulting predictive model, particularly when the quantitative outcome variable has low variability. In such

**Table 4. Explanatory variables for predictive model of the number of railway accidents.**

| Variable | Description | Source |
|---|---|---|
| Year | Year of an accident | Database of railway accidents by Railways of the Slovak Republic |
| Month | Month of an accident | Database of railway accidents by Railways of the Slovak Republic |
| COVID | A dummy indicator with a value of 1 for the months belonging to the COVID-19 pandemic period (in Slovakia from March 2020 till February 2022) and 0 for other months of the period under review | own elaboration |
| Region | Region where an accident happened | Database of railway accidents by Railways of the Slovak Republic |
| Unemployment rate region | Monthly unemployment rate in a region in percentage | Central Office of Labour, Social Affairs and Family of the Slovak Republic |
| Subsistence minimum | Subsistence minimum for an adult person, given based on the growth rate of the cost of living of low-income households Usually change on 1st july each year | Statistical Office of the Slovak Republic |
| Interest rates | Interest rates of national central in percentages per annum | Statistical Office of the Slovak Republic |
| Gross minimum wage | Gross minimum wage, given quarterly in € | Eurostat |
| Inflation | Annual inflation rate provided by the Statistical Office of the SR in percentages | Statistical Office of the Slovak Republic |
| Corruption index | Perceived level of public sector corruption on a scale of 0 (highest corruption) to 100 (no corruption), provided yearly | The Transparency International Agency |
| The number of proposals for divorce men | Cumulative number of proposals for divorce made by men yearly | Statistical Office of the Slovak Republic countryeconomy.com |
| The number of proposals for divorce women | Cumulative number of proposals for divorce made by women yearly | Statistical Office of the Slovak Republic countryeconomy.com |
| The number of divorced marriages | Cumulative number of divorced marriages yearly | Statistical Office of the Slovak Republic countryeconomy.com |
| Fertility rate | Average number of children per woman | Statistical Office of the Slovak Republic countryeconomy.com |
| Life expectancy | Life expectancy at birth, given as the average number of years that a population lives – an indicator of a life quality | Statistical Office of the Slovak Republic countryeconomy.com |
| Divorce rate | Number of divorces per 1,000 population per year | Statistical Office of the Slovak Republic countryeconomy.com |
| At-risk poverty index | Share of people with an equivalised disposable income (after social transfer) below the at-risk-of-poverty threshold, which is set at 60% of the national median equivalised disposable income after social transfers | Statistical Office of the Slovak Republic countryeconomy.com |

**Table 5. Statistical characteristics of quantitative explanatory variables.**

| Variable/ Statistical characteristic | Minimum | Maximum | Mean | Median | Std. deviation | Coef. of Variation (%) |
|---|---|---|---|---|---|---|
| At-risk poverty index | 11.40 | 13.70 | 12.42 | 12.30 | 0.64 | 5.19 |
| Corruption index | 49.00 | 53.00 | 50.80 | 50.50 | 1.22 | 2.41 |
| Divorce proposal men | 3210.00 | 3785.00 | 3515.50 | 3642.00 | 217.04 | 6.17 |
| Divorce proposal women | 5396.00 | 6533.00 | 6070.20 | 6363.00 | 461.14 | 7.60 |
| Divorce rate | 1.50 | 1.80 | 1.67 | 1.70 | 0.13 | 7.92 |
| Fertility rate | 1.40 | 1.63 | 1.53 | 1.54 | 0.07 | 4.25 |
| Gross min wage (€) | 380.00 | 646.00 | 507.55 | 480.00 | 94.77 | 18.67 |
| Inflation (%) | −0.90 | 13.40 | 3.01 | 2.00 | 3.88 | 128.69 |
| Interest rates (%) | 0.99 | 5.66 | 3.43 | 3.37 | 1.05 | 30.75 |
| Life expectancy | 74.60 | 77.80 | 76.98 | 77.30 | 0.84 | 1.10 |
| Marriage rate | 4.40 | 5.80 | 5.23 | 5.40 | 0.46 | 8.74 |
| Min wage (€) | 380.00 | 646.00 | 507.55 | 480.00 | 94.77 | 18.67 |
| No. Of divorced marriages | 8131.00 | 9786.00 | 9060.36 | 9466.00 | 660.81 | 7.29 |
| Subsistence minimum (€) | 198.09 | 234.42 | 207.70 | 205.07 | 10.81 | 5.20 |
| Unemployment rate (%) | 4.88 | 12.39 | 7.28 | 6.67 | 2.09 | 28.67 |
| Unemployment rate region (%) | 2.31 | 17.55 | 6.98 | 5.93 | 3.59 | 51.47 |

**Table 6. Confusion matrix for testing set.**

| Actual accident rate | Predicted accident rate | | | |
|---|---|---|---|---|
| | 1 | 2 | 3 | sum |
| 1 | 24 | 1 | 0 | 25 |
| 2 | 1 | 43 | 4 | 48 |
| 3 | 0 | 1 | 22 | 23 |
| Sum | 25 | 45 | 26 | 96 |

cases, it can be challenging for a model to explain it using the explanatory variables accurately. We determined that three categories for the outcome variable were optimal for achieving a sufficiently high level of model performance. These categories were defined with practical interpretation and usability of the model predictions in mind.

After the categorisation of the outcome variable, the distribution of the values of potential input variables should be depicted by the histograms with respect to the outcome variable categories. Fig 3 presents these distributions.

After creating several models and applying various modelling techniques, we found that the model achieved the best performance when both the outcome variable and all explanatory quantitative variables were categorised. This is also a common step in data mining when creating a predictive model using machine learning techniques. We used an optimal binning method for the individual explanatory variables, with the outcome variable (*accident rate*) as the binning supervisor. The resulting intervals do not have an equal distribution of values or equal width but are instead optimally defined concerning the category of the outcome variable. Quantitative explanatory variables for which optimal binning was not found were omitted from further model creation, as this result indicated that they are not associated with the outcome variable.

These data preparations in the new format represented a step back in the data mining process, which is an example of why these two steps are bidirectionally connected according to CRISP-DM methodology, as shown in Fig 1. After preparing the data for creating the model for railway accident rates, we obtained the variables with the distributions presented in

| Variable | Values | no. of accidents | % share of accidents |
|---|---|---|---|
| year | 2015 | 68 | 12.55 |
| | 2016 | 66 | 12.18 |
| | 2017 | 72 | 13.28 |
| | 2018 | 77 | 14.21 |
| | 2019 | 63 | 11.62 |
| | 2020 | 59 | 10.89 |
| | 2021 | 54 | 9.96 |
| | 2022 | 83 | 15.31 |
| month | 1 | 36 | 6.64 |
| | 2 | 29 | 5.35 |
| | 3 | 47 | 8.67 |
| | 4 | 49 | 9.04 |
| | 5 | 57 | 10.52 |
| | 6 | 41 | 7.56 |
| | 7 | 40 | 7.38 |
| | 8 | 47 | 8.67 |
| | 9 | 47 | 8.67 |
| | 10 | 47 | 8.67 |
| | 11 | 64 | 11.81 |
| | 12 | 38 | 7.01 |
| region | BA | 102 | 18.82 |
| | BB | 43 | 7.93 |
| | KE | 83 | 15.31 |
| | NR | 47 | 8.67 |
| | PO | 72 | 13.28 |
| | TN | 63 | 11.62 |
| | TT | 57 | 10.52 |
| | ZA | 75 | 13.84 |
| COVID | 0 | 426 | 78.60 |
| | 1 | 116 | 21.40 |
| season | spring | 153 | 28.23 |
| | summer | 128 | 23.62 |
| | autumn | 158 | 29.15 |
| | winter | 103 | 19.00 |

**Fig 2. Distributions of categories of qualitative explanatory variables.**

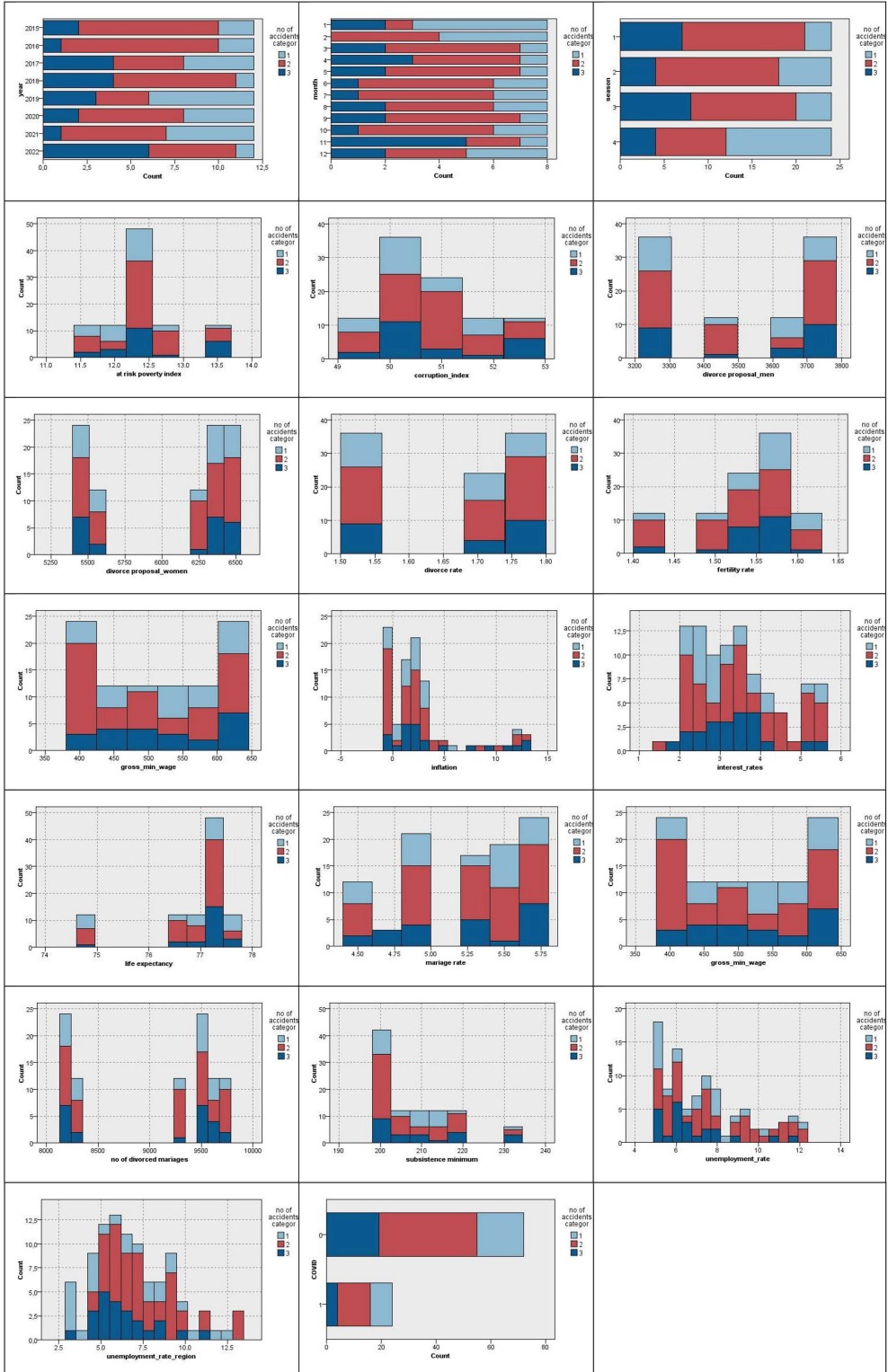

**Fig 3. Distribution of explanatory variables with respect to the outcome variable categories.**

Fig 4. We did not list the variable *interest rate*, as it has almost 50 categories created automatically, each with a low share of cases. We did not adjust this automatically created binning to optimizs it because our tests indicated that doing so would decrease the model's performance.

Finally, we created a prediction model for the number of railway accidents. For this purpose, we tried several machine learning techniques for solving a classification task: logistic regression, classification tree (CART), neural network, and discriminant analysis. The model with the highest performance was created using logistic regression with the forward selection of explanatory variables. The details of this model will be described in the following section of the paper. Here, we briefly characterise the logistic regression method.

Logistic regression is a statistical method used for solving classification tasks. It models the probability of an outcome, which is usually binary. However, it can also be applied when the outcome variable has more than two categories, allowing for the modelling of multiple classes by estimating the probability of each category given the values of predictor (explanatory) variables [39]. Our outcome variable, *accident rate*, has three categories; therefore, logistic regression estimates the probability of each level of accident rate in a certain month, given the values of all explanatory variables listed in Table 4.

If we denote the outcome variable by $Y$ and the vector of explanatory variables by $\boldsymbol{X} = (X_1, X_2, \ldots, X_k)$, the logistic regression model for the probability that a given input $\boldsymbol{X}$ belongs to a particular category of $Y$ can be expressed as:

$$P(Y = j | \boldsymbol{X}) = 1 / \left( 1 + exp \left\{ -1 \cdot (\beta_0 + \beta_1 X_1 + \beta_2 X_2 + \ldots + \beta_k X_k) \right\} \right)$$

where

$P(Y = j | \boldsymbol{X})$ is the probability that the outcome variable (*accident rate*) equals $j \in \{1, 2, 3\}$ given the predictors $\boldsymbol{X}$

$\beta_0$ is the intercept term, its estimated value is $b_0$

$\beta_1, \beta_2, \ldots, \beta_k$ are the coefficients for the predictor (explanatory) variables $X_1, X_2, \ldots, X_k$; their estimations would be denoted by $b_1, b_2, \ldots, b_k$

To estimate the parameters $\beta_0, \beta_1, \ldots, \beta_k$, the maximum likelihood estimation (MLE) method is commonly used. The likelihood function is given by

$$L(\beta_0, \beta_1, \ldots, \beta_k) = \prod_{i=1}^{n} P(Y_i = j | \boldsymbol{X}_i)^{y_i}$$

where

$y_i$ is the observed value of the outcome variable for the $i$-th observation

$X_i$ represents the vector of explanatory variables for the $i$-th observation

$P(Y_i = j | \boldsymbol{X}_i)$ is the probability that the outcome variable $Y$ equals $j \in \{1, 2, 3\}$ for the $i$-th observation, given the explanatory variables $\boldsymbol{X}_i$

Maximising this likelihood function provides estimates $b_0, b_1, \ldots, b_k$ of the model coefficients that best fit the data.

We consider the main advantage of logistic regression to be its interpretability and the readability of the model itself. In comparison with several machine learning models that are considered black boxes, logistic regression allows for the interpretation of the coefficients in the model. This provides an intuitive understanding of the relationship between the explanatory variables and the probability of the outcome.

For a quantitative continuous explanatory variable, each coefficient represents the change in the log odds of the outcome for a one-unit change in the explanatory variable, holding all other variables constant. For categorical explanatory variables, the coefficients represent the change in the log odds of the outcome relative to the reference category of the outcome variable. The log odds is the natural logarithm of the odds, where the odds in the multinomial logistic regression

| Variable | Values | Bin | % share of accidents |
|---|---|---|---|
| subsistence minimum_cat | 1 | ( ; 199.5) | 29.84 |
| | 2 | <199.5 ; 205.1) | 14.92 |
| | 3 | <205.1 ; 210.2) | 13.87 |
| | 4 | <210.2 ; 214.8) | 11.26 |
| | 5 | <214.8 ; 218.1) | 10.21 |
| | 6 | <218.1 ; 234.4) | 12.57 |
| | 7 | <234.4 ; ) | 7.33 |
| gross_min_wage_cat | 1 | ( ; 435.0) | 23.82 |
| | 2 | <435.0 ; 520.0) | 27.75 |
| | 3 | <520.0 ; 646.0) | 34.82 |
| | 4 | <646.0 ; ) | 13.61 |
| inflation_cat | 1 | ( ; - 0.4) | 9.16 |
| | 2 | <- 0.4 ; - 0.2) | 6.28 |
| | 3 | <- 0.2 ; - 0.1) | 5.24 |
| | 4 | <- 0.1 ; 0.7) | 3.14 |
| | 5 | <0.7 ; 0.8) | 1.05 |
| | 6 | <0.8 ; 1.1) | 3.93 |
| | 7 | <1.1 ; 1.4) | 2.88 |
| | 8 | <1.4 ; 1.5) | 4.45 |
| | 9 | <1.5 ; 1.6) | 1.57 |
| | 10 | <1.6 ; 1.7) | 1.57 |
| | 11 | <1.7 ; 1.9) | 7.33 |
| | 12 | <1.9 ; 12.4) | 47.64 |
| | 13 | <12.4 ; ) | 5.76 |
| corruption_index_cat | 1 | ( ; 51.0) | 52.36 |
| | 2 | <51.0 ; 53.0) | 34.03 |
| | 3 | <53.0 ; ) | 13.61 |
| fertility rate_cat | 1 | ( ; 1.4) | 1.83 |
| | 2 | <1.4 ; 1.48) | 10.21 |
| | 3 | <1.48 ; 1.52) | 11.78 |
| | 4 | <1.52 ; 1.54) | 13.61 |
| | 5 | <1.54 ; 1.6) | 52.36 |
| | 6 | <1.6 ; ) | 10.21 |
| life expectancy_cat | 1 | ( ; 77.2) | 33.25 |
| | 2 | <77.2 ; 77.3) | 13.61 |
| | 3 | <77.3 ; 77.4) | 26.18 |
| | 4 | <77.4 ; 77.8) | 13.87 |
| | 5 | <77.8 ; ) | 13.09 |
| divorce rate_cat | 1 | ( ; 1.7) | 35.34 |
| | 2 | <1.7 ; 1.8) | 25.39 |
| | 3 | <1.8 ; ) | 39.27 |
| mariage rate_cat | 1 | ( ; 4.8) | 11.52 |
| | 2 | <4.8 ; 5.4) | 35.34 |
| | 3 | <5.4 ; 5.5) | 13.09 |
| | 4 | <5.5 ; 5.7) | 12.30 |
| | 5 | <5.7 ; 5.8) | 14.14 |
| | 6 | <5.8 ; ) | 13.61 |
| at risk poverty index_cat | 1 | ( ; 11.9) | 11.52 |
| | 2 | <11.9 ; 12.2) | 13.61 |
| | 3 | <12.2 ; 12.3) | 13.61 |
| | 4 | <12.3 ; 12.4) | 21.73 |
| | 5 | <12.4 ; 12.7) | 14.14 |
| | 6 | <12.7 ; 13.7) | 13.35 |
| | 7 | <13.7 ; ) | 12.04 |

**Fig 4. Categorised explanatory variables.**

are defined for a particular outcome category $j \in \{1, 2, 3\}$ as the probability of that category occurring, divided by the probability of the reference category (usually the first category):

$$Odds = \frac{P(Y = j | \boldsymbol{X})}{P(Y = 1 | \boldsymbol{X})}$$

Thus, a positive coefficient indicates an increase in the log odds, and hence the probability, of the outcome relative to the reference category, while a negative coefficient indicates a decrease [39]. The multinomial logistic regression model estimates a separate set of coefficients for each outcome category relative to the reference category, allowing for the prediction of all categories simultaneously.

Another advantage of the logistic regression to be mentioned is its efficiency with large datasets. It is computationally efficient and can handle large datasets with many predictors, making it suitable for big data applications [Menard, 2020] [40]. Moreover, logistic regression provides estimates for the probability of each outcome, which can be particularly useful for decision-making processes in real applications [41]. Additionally, logistic regression is less sensitive to outliers in the explanatory variables compared to other methods [39].

After the *Modelling* phase, the next step of the data-mining process is the *Evaluation* of the model. For this purpose, we used several evaluation measures, all based on the confusion matrix. In our case, this matrix is a 3×3 table table that outlines the correctly and incorrectly predicted cases. Each row of the matrix represents the cases in a predicted category of the outcome variable, while each column represents the cases in the actual category. For a three-category outcome variable, such as our *accident rate* variable, let's denote the categories in general as $C_1, C_2, C_3$. Then, the confusion matrix would then be given by

$$\begin{bmatrix} TP_{C_1} & FP_{C_2} & FP_{C_3} \\ FN_{C_1} & TP_{C_2} & FP_{C_3} \\ FP_{C_1} & FN_{C_2} & TP_{C_3} \end{bmatrix}$$

Each cell of the confusion matrix represents the count of cases where the model's predictions match or mismatch the true category of the outcome across all categories. True positives (TP) are the cases where the predicted category matches the actual one. For instance, if an observation belongs to category $C_1$ and is also predicted as $C_1$ it is considered a true positive for $C_1$, denoted by $TP_{C_1}$. False positives (FP) are those observations where the predicted category does not match the actual. For example, if an observation belongs to category $C_2$ but is predicted as $C_1$, it is considered a false positive for $C_1$ and denoted by $FP_{C_1}$.

True negatives (TN) are those cases where the prediction of not belonging to a specific category is correxct. However, in a multi-class problem, true negatives are usually not used, as they represent the correct rejection of specific categories. Finally, false negatives (FN) are cases incorrectly predicted not to belong to a specific category. For example, if an observation belongs to category $C_1$ but is predicted not to be in $C_1$, it is counted as a false negative for $C_1$, denoted by $FN_{C_1}$.

The evaluation metrics used for measuring the performance of the logistic prediction models are as follows. The evaluation metrics are derived from the confusion matrix, with the most commonly used being the following [Forman, 2023; 42,43,44]:

$$Acc = \frac{TP + TN}{TP + FP + TN + FN}.$$

$$Sen = \frac{TP}{TP + FN}$$

$$Spec = \frac{TN}{FP + TN}$$

$$P = \frac{TP}{TP + FP}$$

where

*Acc* is the overall accuracy of the prediction model, measuring the proportion of correctly classified cases in all categories

*Sen* is sensitivity, representing the proportion of true positive predictions out of the actual positive cases

*Spec* is specificity, representing the proportion of true negative predictions out of the actual negative cases

*P* is precision, representing the proportion of true positive predictions out of all positive predictions.

For a multinomial outcome variable, evaluation metrics must be aggregated by the macro or micro average to summarise the model's performance across multiple categories. In macro averaging, the evaluation metrics are calculated separately for each category, and their average across all categories is taken. This approach is useful when all categories of the outcome variable should be considered equally important. A weighted macro average also considers the size of the specific category and uses the weighted average. On the other hand, micro-averaging aggregates the contributions of all categories before computing the overall evaluation metrics. It sums the correctly and incorrectly predicted cases into one four-field confusion matrix and calculates the evaluation metrics according to the abovementioned equations.

Finally, the prediction performance of the model will be evaluated using the ROC (Receiver Operating Characteristic) curves. The ROC curve is a graphical tool used to assess the predictive performance of a classification model by plotting the TP rate against the FP rate at various threshold levels. The closer the ROC curve is to the top-left corner of the plot, the better the model's performance, as this indicates a higher true positive rate and a lower false positive rate, maximising both sensitivity and specificity.The results of the *Evaluation* phase will be listed in the following section. The model's predictions will be evaluated on the testing part of the data. This means the data will be split into two parts before modelling: a training part will serve for model creation, and the remaining testing part will be used for evaluating the model's performance on previously unknown cases. We used a ratio of 75% for the training part and 25% for the testing part.

Moreover, the resulting model will be evaluated using pseudo-variability statistics, such as Nagelkerke R-square, which measures the proportion of variability in the outcome variable explained by the model. Aditionally, statistics such as $-2\log Likelihood$ should be used for comparing several models, where a lower value indicates a higher quality model. However, it is important to mention that these metrics describe the model created on the training data; therefore, they should suffer from overestimation. For this reason, the other evaluation metrics mentioned above will be presented only for the testing data, to describe the model's performance more accurately.

It is important to mention that evaluating the statistical significance of the explanatory variables used in the prediction model is unnecessary in data mining. If the model is used for predictions, it is not necessarily necessary to check whether all the variables used are statistically significant. The focus is instead on the model's performance in its predictions. Therefore, we do not interpret all the model details, only some interesting aspects.

According to the CRISP-DM methodology, the final step of the data-mining process is the *Deployment* phase, where the prediction model is used in practice and, after some time of its implementation, its performance is re-evaluated. This phase also involves the possible re-estimation of the model if its performance significantly decreases or if the conditions under which the model was originally estimated change dramatically. In this paper, we do not address the proposal of model deployment. However, in the last section of the article, we list several practical consequences of the analysis conducted in this study.

## 6. Results

In this section, we present the results of the modelling and evaluation phases for predicting the number of railway accidents.

To create the prediction model for accident rate, we employed the logistic regression with a forward stepwise selection of explanatory variables. The resulting model, finalised in the fourth step, includes four variables identified as the most important predictors of accident rates. Fig 5 illustrates the importance of these predictors in the final model.

The importance of predictors is calculated such that their cumulative sum totals one. Among the explanatory factors influencing accident rates, *interest rate* emerges as the most critical (importance = 0.65). The second place of importance belongs to the *marriage rate* (importance = 0.19), while the seventh month (*July*) and *fertility rate* account for smaller shares of importance (0.08 and 0.07, respectively). The regression model for accident rates 2 and 3 is detailed in S1 File, with the first category of accident rate 1 serving as the reference.

Examining the regression coefficients reveals that all interest rates carry negative coefficients for accident rates 2 and 3 (the categories with higher and the highest accident frequencies). Approximately one-third of these coefficients are negative, suggesting a lower probability of accidents. Conversely, positive coefficients indicate a higher probability of accident rates for category 3. Therefore, focusing on this high-risk category suggests that higher interest rates increase the probability of the highest railway accident rates. This observation holds true, particularly for interest rates with higher indices, which exhibit the highest coefficients.

Besides that, July (the seventh month) has a noticeable impact by decreasing the probability of the highest railway accident rate compared to other months, holding all other variables constant. A similar effect can be observed in the middle accident rate of category 2, albeit with smaller coefficient values. On the other hand, the marriage rate tends to increase the probability of the highest accident rate in the first two categories (for category details, see Table 7) relative to the last reference category with the highest marriage rate and other variables held constant. Only the middle marriage rate category of accident rate 3 shows a negative impact on the probability of the highest accident rate compared to the highest marriage rate category.

Interestingly, the fertility rate of category 4 increases the probability of the highest accident rate compared to the highest fertility rate category, with a similar effect observed but with a smaller coefficient value for category 5 of the fertility rate.

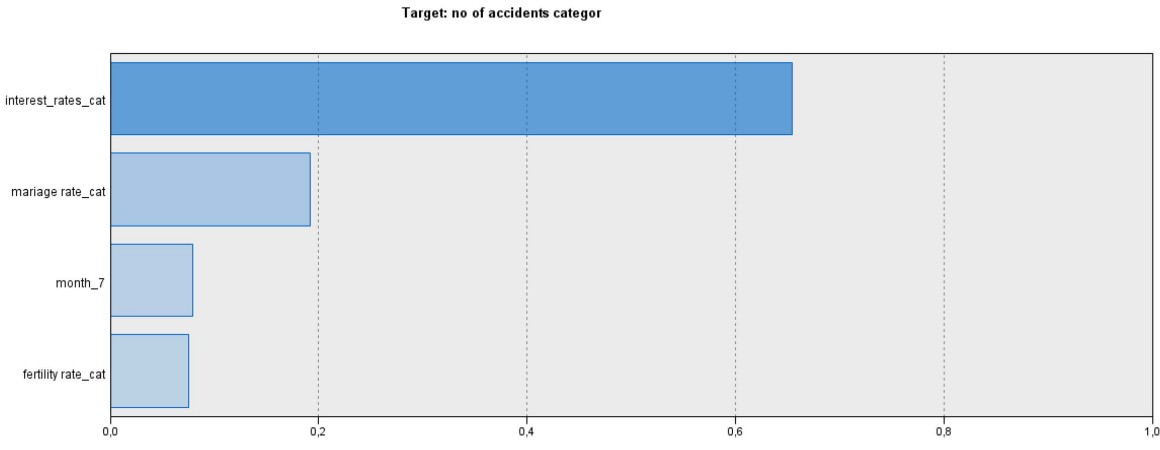

**Fig 5. Predictors importance in the logistic regression model.**

Conversely, similar coefficients but in reverse order are observed for the second accident rate category and the fertility rate. When comparing the coefficients for the marriage rate between the second (middle) and the third (highest) accident rate, it is noteworthy that coefficients are predominantly negative for the middle accident rate, whereas they are mostly positive for the highest accident rate. This implies that with respect to the marriage rate, there is a higher risk of the highest accident rate compared to the lowest accident rate. Conversely, considering the marriage rate, there is a lower risk for the middle accident rate than for the lowest accident rate.

Regarding the quality of the resulting regression model and its predictive power, it is evident that the explanatory power of the model is very high. The Nagelkerke R-square value of 0.964 indicates that the model explains more than 96% of the pseudo-variability in the outcome variable *accident rate*. Similarly, other metrics, such as Cox and Snell's pseudo R-square (0.843) and McFadden's R-square (0.892), also demonstrate the high explanatory power of the model.

The $-2\log Likelihood$ statistics decreased significantly from nearly 200 for the initial model with only the intercept to 21.5 for the fourth model with four explanatory variables. This reduction underscores the improved fit and quality of the model as more variables were included. To avoid confusion, it is important to mention that these metrics describe the model created on the training data, not the testing one. For this reason, for further evaluation of the model's predictive performance, we utilised the confusion matrix detailed in Table 6. Here, the rows represent the actual value of the accident rate, while the columns are the predictions. Correctly predicted cases align along the diagonal, while off-diagonal cells indicate incorrect predictions. We focus solely on results from the testing dataset, as they provide a clearer depiction of the model's performance.

The overall accuracy of our prediction model is 92.7%. Further evaluation measures include computing macro and micro averages from partial confusion matrices for individual categories of accident rates. We calculated sensitivity, precision and accuracy for each category and averaged them using macro average and weighted macro average, where weights are based on the number of cases in each category of the accident rate. Table 7 presents the results, where all the partial confusion matrices are summed into one to calculate micro-averaged evaluation measures.

Our prediction model demonstrates high performance across all categories, with average evaluation measures reflecting strong performance levels. According to the (weighted) macro average, the model achieves an overall accuracy of over 94%. Specifically, the sensitivity for predicting the highest accident rate (category 3) exceeds 95%, indicating that if an accident rate is the highest in a specific month, our model correctly predicts it in over 95% of cases. The precision for this category is over 84%, meaning that if the model predicts the highest accident rate, these predictions would be accurate in over 84% of cases. This slight margin of error in predictions suggests it is preferable to predict an accident unnecessarily than to miss predicting it when it could potentially occur in reality.

Fig 6 illustrates the ROC curves (blue curve) for each category of accident rate (left side – rate 1, middle – rate 2, right side – rate 3). Since the outcome variable was not binary, the ROC must be depicted for each category individually. All the curves lie close to the top left corner, demonstrating the fairy high prediction performance of the model for the railway accident rate.

**Table 7. Individual and average evaluation measures in percentage.**

| Evaluation measure | Accident rate category | | | Average measure | | |
|---|---|---|---|---|---|---|
| | 1 | 2 | 3 | Macro average | Weighted macro average | Micro average |
| **Sensitivity** | 96.00 | 89.58 | 95.65 | 93.75 | 92.71 | 92.71 |
| **Precision** | 96.00 | 95.56 | 84.62 | 92.06 | 95.69 | 92.71 |
| **Accuracy** | 97.92 | 92.71 | 94.79 | 95.14 | 94.56 | 95.14 |

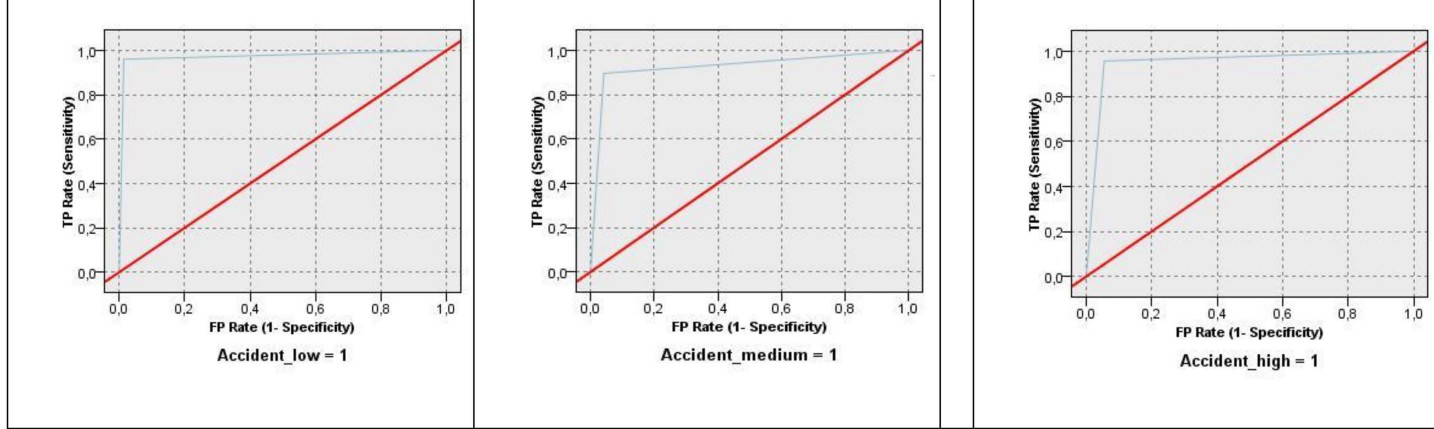

**Fig 6. ROC curves for individual accident rate categories.**

## 7. Discussion

The predictive modelling and statistical analysis presented in this study highlight the significant impact of societal factors on railway accident rates. Using logistic regression with forward stepwise selection, our model identified four key predictors: interest rate as the most significant predictor in the model, marriage rate as the second most influential factor, the month of July on the third place of the significance, and fertility rate. These variables collectively demonstrated the high explanatory power of the prediction model, explaining over 96% of the explained pseudo-variability in accident rates.

The month of July was found to decrease the probability of higher accident rates, reflecting seasonal variations influenced by factors such as weather conditions, holidays, and reduced operational stress. Seasonal variations have been observed in various types of accidents, including those on railways. The month of July, characterised by increased travel and outdoor activities due to favourable weather conditions and variations in social behaviour, could influence accident rates. Ajdacic-Gross et al. [45] and Woo et al. [46] focused their studies on seasonal variations of suicides of various types and documented seasonal peaks in suicide rates, suggesting that similar patterns might exist for other types of incidents, including railway accidents.

Higher interest rates can impact economic conditions, influencing individual stress levels and behaviours. Their association with increased probabilities of higher accident rates may be attributed to their broader economic and psychological effects. While direct studies linking interest rates to railway accidents are limited, the broader relationship between economic stressors and accidents suggests an indirect effect [47,48]. Rising interest rates can restrict access to homeownership, particularly for young individuals and families, leading to increased uncertainty and frustration about the future. Furthermore, for individuals who already have mortgages or other loans, higher interest rates result in increased monthly payments. This can create substantial financial strain, particularly for those already facing economic uncertainty or tight family budgets. This added burden can contribute to increased psychological distress, depression, and suicidal behaviour, potentially contributing to the observed association with the increased number of railway suicides.

Regarding the marriage rate, our findings suggest that higher marriage rates are associated with an increased likelihood of lower accident rates. This finding is consistent with existing literature indicating that marriage provides social integration, emotional support, and economic stability, which may serve as a protective factor against suicidal behaviour [49,50]. Research indicates that divorced or separated individuals have higher suicide rates compared to their married counterparts [51,52]. A stable societal environment could reduce negative feelings, which may explain the relationship between marriage rates and the lower incidence of suicides and suicide attempts on railway tracks.

Despite being the least significant of the four factors, the fertility rate still demonstrated a notable influence. While specific studies linking fertility rate to railway accidents are rate, the association between fertility rates and societal well-being is well-documented [53,54]. Higher fertility rates generally tended to be associated with increased accident rates compared to the highest fertility rate category. One potential explanation for the relationship between fertility rates and railway accident rates is that higher fertility rates may indicate stronger family structures, which could provide protective effects similar to those observed with marriage rates [55]. Additionally, demographic and economic conditions associated with varying fertility rates, such as employment stability, social policies supporting families, and overall economic optimism, could indirectly influence mental health outcomes and, thus, the risk of suicide.

Interestingly, our study did not reveal the COVID-19 pandemic period as a significant factor influencing the rate of suicide-related railway accidents. These results partially align with observations from several other studies that examined suicide trends during the pandemic. For instance, a study by Pirkis et al. (2021) [56] reported that the number of suicides remained largely unchanged or even declined during the initial months of the pandemic compared to expected levels based on the pre-pandemic period. However, this study focused solely on high-income and upper-middle-income countries. In contrast, a study by Dube et al. [57] revealed that the COVID-19 pandemic led to an increase in suicidal behaviour, particularly among the most vulnerable groups, such as younger individuals, women, and those from democratic countries. In our study, the average number of railway accidents during the pre-pandemic and pandemic periods was comparable, which was likely a reason why this variable was not statistically significant in the model. Moreover, the periods were included in the model as dummy indicators for individual years, along with the dummy variables for the pandemic. If the pandemic had impacted the accident rate, it would likely have been visible in the model either by including the pandemic variable or through those for individual years of the period under review as statistically significant variables.

Other similar studies have also employed multinomial logistic regression to predict specific aspects of railway accidents. Wang et al. [1] developed a model to predict the types of railway accidents involving foreign objects in railway encroachment. Their study identifies factors and characteristics of trespasser injury and assesses the risk level associated with illegal trespassing incidents. Hu et al. [58] used multinomial logistic regression to predict railway accidents, considering 35 features related to railway infrastructure, highway crossing, traffic and other variables to explain accident severity across four grades.

In addition to logistic regression, various machine learning techniques have been utilised for predicting railway accidents. Chang and Gong [59] employed text mining to analyse accident reports and extract crucial factors contributing to railway accidents. Bridgelall and Tolliver [60] identified the extreme gradient boosting technique as the most effective in predicting accident types. The authors used scoring techniques to identify key explanatory factors. Zheng et al. [61] utilised decision trees to model highway-rail grade crossing crashes, highlighting significant factors contributing to these rare but often catastrophic crashes.

Our study focuses on explaining railway accidents categorised as suicides or suicide attempts using socio-economic factors, distinguishing it from previously published studies that predominantly attribute accidents to the railway or traffic-related factors. Our model's coefficients provide clear insights into the relationship between predictor variables and the likelihood of different accident rates, enabling better understanding and decision-making processes in railway safety management.

The findings from our logistic regression model, designed to predict the railway accident rate, highlight several key advantages. Firstly, logistic regression proves advantageous due to its interpretability and efficiency. The model's coefficients offer clear insights into the relationship between explanatory variables and the likelihood of different accident rates, enabling better understanding and decision-making processes in railway safety management. Moreover, employing the CRISP-DM methodology ensures the model's robustness and adaptability. This structured approach to data mining allows for easy application and adjustment as conditions of explanatory variables evolve. Such flexibility is crucial for maintaining the model's relevance and accuracy over time, thereby enhancing railway safety management by enabling proactive risk mitigation measures.

However, it is essential to acknowledge the limitations and potential areas for improvement in future research. While our study provides valuable insights, it is not without limitations. Firstly, we focused on a limited number of socio-economic factors, whose selection was guided by existing literature linking macroeconomic conditions to suicide risk to explore their influences on railway accidents of suicide type. Nevertheless, the main reason for selecting the explanatory variables for our study was the data availability across the period under review, ensuring consistency and comparability of our results. While other potential explanatory factors, such as cultural attitudes towards safety, mental health conditions, or technological advancements in railway systems, may also play a role, reliable and longitudinal data on these aspects in the Slovak Republic were not available for inclusion. Additionally, the macroeconomic indicators chosen offer a broader structural perspective, making them particularly suitable for analysing general impact rather than individual-level behavioural factors. Future research in this area could expand on our findings by integrating additional variables as more comprehensive datasets become accessible.

Furthermore, our outcome variable – the monthly accident rate of railway suicides and suicide attempts – was derived from aggregated numbers of accidents, which did not allow for the inclusion of microdata such as education level, gender, or age of the accident victims. This aggregation, while necessary for statistical modelling, limits the ability to analyse demographic risk factors and individual-level psychological determinants of suicide. However, our study primarily focused not on the accident from a psychological perspective but rather from the viewpoint of railway safety. Moreover, the majority of data used in the analysis were aggregated at the national level, which may obscure regional differences and local factors influencing railway safety. Therefore, future research should consider more granular, region-specific data where available to capture these local nuances better. Additionally, as more detailed datasets become accessible, studies could explore the role of mental health services, crisis intervention programmes, and broader social determinants of suicide in railway environments. Another possible direction is to focus future work on incorporating more sophisticated techniques, such as ensemble methods, to potentially enhance the predictive performance and robustness of the predictive model.

Anyway, the issue of this type of rail suicide and its prevention is highly complex. Several particularly effective measures can help reduce the number of suicides on the railways. These measures typically involve a combination of prevention, awareness-raising, technical solutions, and mental health support. Some examples to suggest to the responsible individuals are as follows. Firstly, data on suicides and attempted suicides can be utilised to identify areas and times of highest risk. Based on this data, railway companies can optimise their response and concentrate on areas that require immediate attention. Secondly, the most critical points should be monitored by cameras and sensors, immediately identifying individuals in danger and allowing for immediate reaction. Thirdly, there are examples from other countries of installing protective barriers at critical points in railway infrastructure. These barriers can prevent people from entering the tracks. All three measures mentioned so far are strongly connected to the analyses of historical records of railway incidents to identify the most critical points on the railways or periods with the highest risk. Therefore, analytical studies, such as those presented in our paper, are of high importance, allowing for employing evidence-based protective measures.

Another viewpoint on the protective measures is providing mental health support and intervention. This topic is of great importance and should be a priority for the government to support the organisations providing mental health support, consultancy, and crisis lines. Critical locations on rail infrastructure can be equipped with tables displaying emergency contact telephone numbers.

Last but not least, based on an analysis of critical locations and times of accidents, it is possible to identify an increased likelihood of an accident situation and alert railway staff. Employees such as drivers, conductors and station staff should be trained to identify risk situations and provide first aid in crisis situations. They should also know how to communicate effectively with individuals in psychological distress and guide them to the appropriate help.

In any case, all these measures require long-term cooperation between railway companies, organisations providing mental health support, and policymakers.

## 8. Conclusion

This study contributes to the expanding literature on railway safety by emphasising the critical role of societal factors in accident prediction. It showcases the application of logistic regression as a robust method for forecasting railway accidents, highlighting its strengths in terms of interpretability, efficiency, and practical utility. Leveraging the CRISP-DM methodology ensures that the model is not only easy to apply but also adaptable to changes in explanatory variables, maintaining its relevance and accuracy over time.

Identifying the key socio-economic factors with the highest explanatory power in predicting railway accident rates, our study highlights interest rate, marriage rate, fertility rate and month of July. Interest rates emerged as the most significant predictor in our model, indicating that higher interest rates are generally associated with increased probabilities of higher accident rates. The marriage rate, the second most influential factor, exhibited varying impacts across different accident rate categories. Higher marriage rates were associated with an increased probability of lower accident rates, suggesting a more stable societal environment where fewer disruptions.

The evaluation metrics, derived from the confusion matrix, demonstrate the model's robustness. By using macro and micro averages, we ensured a comprehensive assessment across all categories of the outcome variable.

Overall, our study contributes to the growing body of literature on railway safety by suggesting ways to improve safety measures and potentially aiding suicide prevention efforts. The high explanatory power of our model highlights the importance of considering broad societal factors in railway safety management and planning. The practical applications of our model are promising, offering a data-driven approach to railway safety management. Implementing such models could help identify high-risk periods and conditions, allowing for proactive measures to mitigate accident risks. In general, policies promoting economic growth can indirectly enhance railway safety by increasing the resources available for maintenance and safety improvements. Furthermore, targeted interventions in urban areas, such as advanced automated control systems, are essential to manage these regions' higher risk of accidents. The model's ability to predict accident rates accurately underscores its potential as a valuable tool for railway safety practitioners.

To conclude, as the transportation sector evolves, leveraging analytical tools for its management will be key to ensuring safer and more efficient railway operations. The study confirms the effectiveness of logistic regression, but on the other hand, the importance of high-quality input data must also be mentioned. Enhancing data quality, addressing potential biases, and exploring advanced predictive techniques will be crucial for further advancing this field.

## Supporting information

**S1 File. Annex A: Logistic regression model.**
(DOCX)

## Author contributions

**Conceptualization:** Jaroslav Mašek.

**Data curation:** Juraj Čamaj, Lucia Duricova.

**Formal analysis:** Juraj Čamaj, Lucia Duricova.

**Investigation:** Jaroslav Mašek.

**Methodology:** Lucia Duricova.

**Project administration:** Jaroslav Mašek.

**Resources:** Jaroslav Mašek, Juraj Čamaj.

**Software:** Lucia Duricova.

**Supervision:** Jaroslav Mašek.

**Validation:** Juraj Čamaj, Lucia Duricova.

**Visualization:** Lucia Duricova.

**Writing – original draft:** Lucia Duricova.

**Writing – review & editing:** Jaroslav Mašek, Lucia Duricova.

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
