## [Decision Letter · Decision Letter 0]

14 Feb 2025

PONE-D-24-42356Analysing Factors Influencing Railway Accidents: A Predictive Approach Using Multinomial Logistic Regression and Data MiningPLOS ONE

Dear Dr. Mašek,

Thank you for submitting your manuscript to PLOS ONE. After careful consideration, we feel that it has merit but does not fully meet PLOS ONE’s publication criteria as it currently stands. Therefore, we invite you to submit a revised version of the manuscript that addresses the points raised during the review process.

We look forward to receiving your revised manuscript.

Kind regards,

Satabdi Mitra, M.D(Community Medicine )

Academic Editor

PLOS ONE

Journal Requirements:

“The paper was supported by the KEGA Agency by Project 010ŽU-4/2023 "Innovative approaches in

teaching in the field of transport focused on railway traffic management, with the support of risk and crisis

management", that is solved at the Faculty of Operations and Economics of Transport and Communication,

University of Žilina.”

Reviewers' comments:

Reviewer's Responses to Questions

**Comments to the Author**

1. Is the manuscript technically sound, and do the data support the conclusions?

Reviewer #1: Yes

Reviewer #2: Yes

Reviewer #3: Yes

2. Has the statistical analysis been performed appropriately and rigorously? 

Reviewer #1: Yes

Reviewer #2: Yes

Reviewer #3: Yes

3. Have the authors made all data underlying the findings in their manuscript fully available?

Reviewer #1: Yes

Reviewer #2: Yes

Reviewer #3: No

4. Is the manuscript presented in an intelligible fashion and written in standard English?

Reviewer #1: Yes

Reviewer #2: Yes

Reviewer #3: Yes

5. Review Comments to the Author

Reviewer #1: 1. Graphical representation needed.

2. Age, sex, demographic data missing.

3. The present study should become a base for further study on prevention of such accidents, so kindly highlight the steps taken by the authors corresponding to railway officials in the discussion part.

Reviewer #2: It was a very good study and the manuscript is well prepared. Following are some of my comments:

1. Was there an attempt to see what proportion of accidents happened due to non compliance to their Railways Act ? (Act No. 513/2009 on Railways) ?

2. Is there any statistically significant impact in relation to Covid ? This information is somehow unclear in the manuscript although it says the numbers were high in the year 2021 and 2022 ?

3. Your study has identified four important predictors, which has impacted the number of accidents: interest rate, marriage rate, the month of July, and fertility rate. For the month of July you have attributed to the following factors - seasonal variations influenced by factors such as weather conditions, holidays, and reduced operational stress. In the same way what could be other probable factors in interest rate, marriage rate and fertility rate which can affect the number of train accidents. It would be nice to look at it as well.

Reviewer #3: Review Comments on Manuscript: “Analysing Factors Influencing Railway Accidents: A Predictive Approach Using Multinomial Logistic Regression and Data Mining”

Thank you for the opportunity to review this manuscript, which develops a logistic regression model to identify the key socio-economic factors with the highest explanatory power in predicting railway accident rates. The study addresses an important issue and provides valuable insights into the roles of interest rates, marriage rates, and fertility rates in correlating with accident probabilities. However, there are several areas where the manuscript could benefit from further clarification and improvement. My detailed comments are as follows:

My detailed comments are as follows:

Introduction

1.Data Clarification

On page 2, the manuscript states: “By leveraging comprehensive data over nearly two decades, the research offers robust insights into accident causation, which is vital for developing effective safety strategies.” However, there seems to be confusion regarding the time period of the data used. Why is it referred to as nearly two decades of data? Isn’t the data from 2015 to 2022? This needs to be clarified to ensure the accuracy of the statement.

Literature Review

2.Depth and Contribution

On page 2, the manuscript mentions: “Beyond technical and organisational factors, societal dynamics play a significant role in railway accidents. Some studies have explored the relationship between various societal factors, such as economic conditions, demographic trends, and mental health indicators, and the occurrence of railway accidents (Kim and Lim, 2024; Read et al., 2012; Cooper et al., 2019).” While societal factors are central to this study, the literature review is somewhat simplistic. It doesn’t sufficiently elaborate on what previous research has done, the findings they’ve uncovered, or how these studies have contributed to the knowledge gap that this study aims to address. It would strengthen the manuscript to explain how past research has informed this study’s focus and what gaps remain in the literature that this research aims to fill.

3.Citation Formatting

On page 2, the in-text citation format is inconsistent. Please ensure that the citation style is uniform throughout the manuscript for clarity and consistency.

Theoretical Background

4.The title “Theoretical Background” might not be appropriate for this section. Typically, a “Theoretical Background” section in a research paper discusses the theories, models, or frameworks that inform the study. However, this section primarily focuses on regulations, legal frameworks, and accident classifications related to railway safety, which is more descriptive than theoretical. In its current form, this section would be more appropriate as a “Regulatory Background” or “Contextual Background”, as it focuses on the legal and regulatory framework surrounding railway safety. A theoretical background section should ideally discuss accident causation theories, risk management models, or socio-economic theories relevant to the study of railway accidents, which is missing here.

Methods

5.Selection of Explanatory Factors and Exclusion of Other Variables

On page 6, Table 4 mentions that the study attempts to use all explanatory factors for the number of accidents across different months and regions of the country. However, the rationale behind selecting these explanatory factors is unclear. Could the authors explain why these specific factors were chosen for the analysis? This would help readers understand the justification for the model and its variables.

In the limitations section of the manuscript, the authors state: “Additionally, this study focuses on a limited number of socio-economic factors; therefore, future studies could explore other potential influences, such as educational levels, cultural attitudes towards safety, or the impact of technological advancements in railway systems.” However, the reasons for excluding these variables in this study are not explained as well. It would be useful to include a discussion on why these other factors were not considered.

Discussion

6.Linking Findings with Previous Research

On page 16, the manuscript identifies the four key explanatory factors with the highest explanatory power: interest rate, marriage rate, fertility rate, and the month of July. However, since these factors have been identified as significant, it would be beneficial to relate them to previous studies. The authors could explore potential reasons why these factors influence accident rates, based on existing literature or theoretical frameworks.

While I understand that the core aim of this study is to propose a robust logistic regression model for forecasting railway accidents, it would still be helpful to provide some context around these factors, explaining why they matter in the context of railway safety.

I hope the authors find my comments helpful in refining and strengthening their manuscript. I wish the authors all the best moving forward.

6. PLOS authors have the option to publish the peer review history of their article (what does this mean? ). If published, this will include your full peer review and any attached files.

**Do you want your identity to be public for this peer review?** For information about this choice, including consent withdrawal, please see our Privacy Policy .

Reviewer #1: **Yes: ** Santhosh Kumar Tumkur Narayanappa

Reviewer #2: No

Reviewer #3: No

---

## [Author Response · Author response to Decision Letter 1]

26 Mar 2025

Dear Reviewers, please see our responses to your comments in the attachment.

---

## [Decision Letter · Decision Letter 1]

22 Aug 2025

PONE-D-24-42356R1Analysing Factors Influencing Railway Accidents: A Predictive Approach Using Multinomial Logistic Regression and Data MiningPLOS ONE

 Dear Dr.  Mašek,

Thank you for submitting your manuscript to PLOS ONE. After careful consideration, we feel that it has merit but does not fully meet PLOS ONE’s publication criteria as it currently stands. Therefore, we invite you to submit a revised version of the manuscript that addresses the points raised during the review process.

 Please submit your revised manuscript by Oct 06 2025 11:59PM. If you will need more time than this to complete your revisions, please reply to this message or contact the journal office at plosone@plos.org . Please include the following items when submitting your revised manuscript:

We look forward to receiving your revised manuscript.

Kind regards,

Hamed Aghaei, Ph.D.

Academic Editor

PLOS ONE

Journal Requirements:

Reviewers' comments:

Reviewer's Responses to Questions

**Comments to the Author**

1. If the authors have adequately addressed your comments raised in a previous round of review and you feel that this manuscript is now acceptable for publication, you may indicate that here to bypass the “Comments to the Author” section, enter your conflict of interest statement in the “Confidential to Editor” section, and submit your "Accept" recommendation.

Reviewer #1: All comments have been addressed

Reviewer #2: All comments have been addressed

Reviewer #3: All comments have been addressed

Reviewer #4: All comments have been addressed

Reviewer #5: All comments have been addressed

2. Is the manuscript technically sound, and do the data support the conclusions?

Reviewer #1: Yes

Reviewer #2: Yes

Reviewer #3: Yes

Reviewer #4: Yes

Reviewer #5: Partly

3. Has the statistical analysis been performed appropriately and rigorously? 

Reviewer #1: Yes

Reviewer #2: Yes

Reviewer #3: Yes

Reviewer #4: Yes

Reviewer #5: No

4. Have the authors made all data underlying the findings in their manuscript fully available?

Reviewer #1: Yes

Reviewer #2: Yes

Reviewer #3: Yes

Reviewer #4: Yes

Reviewer #5: Yes

5. Is the manuscript presented in an intelligible fashion and written in standard English?

Reviewer #1: Yes

Reviewer #2: Yes

Reviewer #3: Yes

Reviewer #4: Yes

Reviewer #5: Yes

6. Review Comments to the Author

Reviewer #1: 1.Railway has categorized the accidents into Classification of Accidents: Accidents are classified under following heads:-

I. Train Accidents.

II. Yard Accidents.

III. Indicative Accidents.

IV. Equipment failures.

V. Unusual incidents.

Suicidal attempt involving railway track is called Rail suicide

The railway accident and rail suicide are dealt differently as per the law of the country and hence I advise the author to look into the matter regarding the title and AIM of the study

Reviewer #2: A good effort to study an important problem which is not very often studied, in spite of its great relevance. The manuscript in the modified form looks fine for publication, from my end. They have employed the novel techniques and I am sure the information that they have brought out in this study will be a stimulus for thinkers and researchers in this area to further explore in this space there by augmenting the railway safety at large.

Reviewer #3: (No Response)

Reviewer #4: (No Response)

Reviewer #5: The comments to the authors have been uploaded in the attachment due to character count limit. Kindly go through the same. Thank you.

This manuscript addresses a critical issue with a methodologically sound and innovative approach. However, major revisions are required to strengthen the epidemiologic interpretation, improve model transparency, and clarify generalisability.

7. PLOS authors have the option to publish the peer review history of their article (what does this mean? ). If published, this will include your full peer review and any attached files.

**Do you want your identity to be public for this peer review?** For information about this choice, including consent withdrawal, please see our Privacy Policy .

Reviewer #1: **Yes: ** Santhosh Kumar Tumkur Narayanappa

Reviewer #2: No

Reviewer #3: No

Reviewer #4: **Yes: ** DR GOPINATH K

Reviewer #5: **Yes: ** Mohammed Shoyaib Khazi

---

## [Author Response · Author response to Decision Letter 2]

17 Sep 2025

Responds to reviewers are uploaded in separately documents.

---

## [Decision Letter · Decision Letter 2]

22 Sep 2025

Analysing Factors Influencing Railway Accidents: A Predictive Approach Using Multinomial Logistic Regression and Data Mining

PONE-D-24-42356R2

Dear Dr. Mašek ,

We’re pleased to inform you that your manuscript has been judged scientifically suitable for publication and will be formally accepted for publication once it meets all outstanding technical requirements.

Kind regards,

Hamed Aghaei, Ph.D.

Academic Editor

PLOS ONE

Additional Editor Comments (optional):

Reviewer #5:

Reviewers' comments:

Reviewer's Responses to Questions

**Comments to the Author**

1. If the authors have adequately addressed your comments raised in a previous round of review and you feel that this manuscript is now acceptable for publication, you may indicate that here to bypass the “Comments to the Author” section, enter your conflict of interest statement in the “Confidential to Editor” section, and submit your "Accept" recommendation.

Reviewer #5: All comments have been addressed

2. Is the manuscript technically sound, and do the data support the conclusions?

Reviewer #5: Yes

3. Has the statistical analysis been performed appropriately and rigorously? 

Reviewer #5: Yes

4. Have the authors made all data underlying the findings in their manuscript fully available?

Reviewer #5: Yes

5. Is the manuscript presented in an intelligible fashion and written in standard English?

Reviewer #5: Yes

6. Review Comments to the Author

Reviewer #5: I have carefully examined the authors’ responses to my previous comments.

Adjusted Odds Ratios and Confidence Intervals

The authors have added 95% confidence intervals and reorganised the regression outputs into a clearer format. This substantially improves the transparency and interpretability of the results.

COVID Variable and Temporal Analysis

The authors have tested for autocorrelation, used the Box–Ljung test, and incorporated dummy variables for months, seasons, and years as temporal predictors. While more advanced time-series modelling could have been considered, the justification for using logistic regression is scientifically sound and acceptable.

Model Performance and Overfitting Concerns

The authors addressed overfitting concerns by reporting evaluation metrics only on the test dataset from a 75:25 split. While cross-validation or bootstrapping would have provided stronger evidence of generalisability, the explanation regarding software constraints is reasonable, and the presented metrics are sufficient to support the model’s performance.

Overall Assessment:

The authors have substantially addressed Reviewer 5’s comments. The revisions improve the clarity, robustness, and interpretability of the manuscript. The only remaining limitation is the lack of cross-validation, but given the constraints and the adequate justification provided, I find the responses satisfactory.

Recommendation:

The authors have adequately addressed my concerns, and I am satisfied with the current revision. Hence, I recommend this manuscript for publication.

7. PLOS authors have the option to publish the peer review history of their article (what does this mean? ). If published, this will include your full peer review and any attached files.

**Do you want your identity to be public for this peer review?** For information about this choice, including consent withdrawal, please see our Privacy Policy .

Reviewer #5: **Yes: ** Dr Mohammed Shoyaib Khazi

---

## [Editor Report · Acceptance letter]

PONE-D-24-42356R2

PLOS ONE

Dear Dr. Mašek,

I'm pleased to inform you that your manuscript has been deemed suitable for publication in PLOS ONE. Congratulations! Your manuscript is now being handed over to our production team.

Kind regards,

on behalf of

Dr. Hamed Aghaei

Academic Editor

PLOS ONE